# Microwave-Assisted Valorization of Tomato Pomace for Pectin Recovery: Improving Yields and Environmental Footprint

**DOI:** 10.3390/foods14091516

**Published:** 2025-04-26

**Authors:** Nikolina Golub, Emerik Galić, Kristina Radić, Nada Smigic, Ilija Djekić, Sandra Pedisić, Dubravka Vitali Čepo

**Affiliations:** 1Department of Nutrition and Dietetics, University of Zagreb Faculty of Pharmacy and Biochemistry, Ante Kovačića 1, 10000 Zagreb, Croatia; nikolina.golub@pharma.unizg.hr (N.G.); emerik.galic@pharma.unizg.hr (E.G.); kristina.radic@pharma.unizg.hr (K.R.); 2Department of Food Safety and Quality Management, University of Belgrade Faculty of Agriculture, Nemanjina Street 6, 11080 Belgrade, Serbia; nadasmigic@agrif.bg.ac.rs (N.S.); idjekic@agrif.bg.ac.rs (I.D.); 3Centre for Food Technology and Biotechnology, University of Zagreb Faculty of Food Technology and Biotechnology, Petra Kasandrića 3, 23000 Zadar, Croatia; sandra.pedisic@pbf.unizg.hr

**Keywords:** pectin, tomato pomace waste, microwave-assisted extraction, mechanical ball milling, life cycle assessment

## Abstract

Rising industrial demands emphasize the need for exploring other non-traditional sources for obtaining pectin. As efforts to enhance circular economy practices and reduce reliance on primary resources intensify; the focus has shifted towards utilizing various types of agricultural and food industry waste; including tomato pomace waste (TPW). In this work; the microwave-assisted extraction (MAE) and TPW pretreatment methods were optimized to improve pectin yields and decrease the environmental impact of the extraction process; compared to conventional solvent extraction (CSE). The response surface methodology was used to model the optimization process. The physico-chemical properties of pectin were determined using titrimetric methods and FTIR spectroscopy. A life cycle assessment (LCA) was applied to assess the environmental impact of MAE and CSE. Optimal microwave conditions (11.66 min/600 W/pH 1) yielded two times more pectin than CSE (2 h/85 °C/pH 1.5). Pre-processing treatments (mechanical ball milling and defatting) showed marginal effects on pectin yields and properties; and; therefore; can be omitted in order to reduce the energy consumption of the process. The LCA showed that single conventional extraction treatment had two times higher values of the ecological footprint compared to MAE; for all indicators. The results suggest that MAE can be used as an efficient green method for pectin extraction from TPW

## 1. Introduction

Pectin is a heteropolysaccharide that serves as a structural component in the cell walls and intercellular layers of plant cells and is particularly abundant in fruits like citrus and apples. It finds extensive use as a gelling agent, stabilizer, emulsifier and thickener in both food products and pharmaceutical formulations. Additionally, it is prescribed as a dietary supplement or therapeutic agent for treating conditions such as diarrhea or throat irritations [1]. Despite its widespread occurrence across various plant species, only pectin derived from specific sources exhibits suitable functional properties for industrial applications. The functionality of pectin is largely contingent upon its capacity to form gels, which is primarily influenced by molecular size and the degree of methoxylation (DM), the characteristics that can vary significantly based on the source material and extraction conditions [2]. Commercially available pectin is categorized into two primary types: low-methoxyl (LM) pectin, which contains less than 50% methyl ester groups and requires calcium ions for gelation, and high-methoxyl (HM) pectin, which contains more than 50% methyl ester groups and requires sugar and acid for gelation. The strength of the gel formed by pectin is primarily correlated with its molecular weight; higher average molecular weights result in increased viscosity of the gel formed, thereby enhancing gel strength.

Industrial pectin is predominantly derived from citrus peels and apple pomace. However, over the past few decades, there has been a notable increase in demand for pectin, with projections indicating continued market growth between 2024 and 2030, projected to reach $1.8 billion by 2026 [3,4]. This escalating demand has sparked a significant interest in exploring non-traditional sources of pectin. In line with efforts to enhance circular economy practices by reducing reliance on primary resources, research focus has shifted towards evaluating the feasibility of utilizing diverse agricultural and food industry waste materials as alternative raw materials for obtaining pectin [5,6]. Tomato pomace waste (TPW) could serve as a valuable pectin source since it contains approximately 10–15% pectin content when dried. This highlights its potential as the secondary raw material for the commercial-scale production of this polysaccharide [7].

Tomato processing industries produce varying amounts of TPW, typically ranging from 2% to 10% of the total weight of processed tomato fruits. This waste primarily consists of peels, seeds, vascular tissues and pulp residue. Based on these figures, the global tomato processing industry generates approximately 6.2 million to 9 million tons of TPW annually [8]. Given the increased demand for pectin and the need to shift towards circular economy concepts, recent investigations into TPW as an alternative source of pectin have focused on developing efficient and sustainable green extraction techniques. These techniques aim to achieve sufficient yields and product quality while minimizing the environmental impact. Current scientific efforts are focused on modeling the extraction parameters that significantly influence the yields and quality of the product, such as solvent type, extraction time, temperature, pressure and solid-to-solvent ratio, with the aim of developing more sustainable processes with reduced production costs, thus making the extraction economically viable [8].

Different green extraction techniques have been optimized for obtaining pectin from TPW such as ultrasound-assisted extraction (UAE), microwave-assisted extraction (MAE), ohmic-assisted extraction (OHAE) and high-pressure extraction (HPE), as well as combined techniques such as ultrasound-assisted microwave extraction (UAME) and ultrasound-assisted ohmic heating extraction (UAOHE). As summarized by Radić and co-workers [8], obtained pectin yields varied significantly, depending on the raw material and applied extraction technique/extraction conditions. The yields obtained from the intact TPW ranged from 4.3% (obtained via conventional solvent extraction (CSE)) [9] up to 35.7%, (obtained through ammonium oxalate/oxalic acid UAE) [10]. Even though green extraction techniques in the majority of available research produced higher yields compared to CSE, extraction conditions such as duration, temperature, or solvent pH, were shown to be the major factors influencing the utilization of the extraction process. In the majority of available investigations, yields were increased by applying higher temperatures, lower pH values and prolonged extraction times, regardless of the extraction technique.

As reviewed recently by Radić and co-workers [8], the major issue concerning the novel extraction procedures is their scale-up ability to pilot- or industrial-scale in terms of technological readiness of the procedure, the costs of scale-up and maintenance, or both. Recent studies indicate that the industrial application of TPW primarily involves its conversion into animal feed, compost, or biogas through anaerobic digestion, while the extraction of valuable compounds such as total carotenoids, polyphenols and pectin still remains in the experimental phase, mainly at the laboratory scale [11]. In addition to known advantages, such as low solvent consumption, short extraction time and increased efficiency, MAE is also considered to be a promising extraction technique due to the satisfactory level of technological readiness and the lower cost of scale-up, compared to other green extraction techniques [12]. Namely, microwave reactors are already available for both the pilot and industrial scales. Available literature data show the great potential of industrial applications of MAE, showing comparable yields of the essential oils from *Peumus boldus* leaves at both laboratory and industrial levels [13] and higher yields of polyphenols from lettuce at the industrial level [14].

Despite the general advantages of MAE as a green extraction technique, its applicability for tomato pectin extraction has not been thoroughly investigated. The only available data are from Lasunon and co-workers [15] and Sengar and co-workers [16], who used a household microwave oven for pectin extraction and achieved satisfying yields (12.0 and 25.4%, respectively). The applicability of laboratory closed vessel systems, which offer numerous advantages compared to household microwaves (such as rapid heating, temperature control and the utilization of temperatures above the solvent’s boiling point to enhance solubility), for MAE has not been investigated so far. Additionally, the importance of raw material pretreatment in the process of obtaining pectin from TPW has been recognized only recently and has not been investigated thoroughly; a study by Van Audenhove and colleagues [17] is the sole investigation into how pretreating raw materials affects pectin yields and properties. They found that using high-pressure treatment before extracting pectin substantially boosted yields. Additionally, the extracted pectin had a lower dry matter content and a higher molecular weight compared to untreated samples. Finally, it has to be emphasized that even though MAE is generally known as the green extraction technique that offers environmental benefits compared to CSE, its actual environmental advantages in pectin extraction have not been investigated so far by any relevant methodology.

Considering all the above, the primary goal of this investigation was to optimize the MAE conditions to maximize the yields of TPW pectin with satisfactory chemical characteristics. To our knowledge, this is the first attempt to develop the procedure using a closed vessel microwave extraction system that could result with higher pectin yields, shorter extraction times and reduced solvent utilization, compared to conventional solvent extraction but also compared to household microwave ovens used for this purpose by other authors [8]. Additionally, this paper investigated for the first time how different TPW pretreatments (drying, milling and defatting) can be optimized to improve pectin yields and decrease the environmental impact of the extraction process. In parallel, to analyze the potential of employing MAE as a “green” extraction solution, a life cycle assessment (LCA) was used to compare the ecological footprint of MAE with CSE by assessing the ecological acceptability of the proposed process. This is the first time, to our knowledge, that an LCA was applied to realistically and objectively assess the environmental advantages of MAE as a “green extraction method” applied for pectin extraction. These goals contribute greatly to the attempts of the reutilization of food industry waste for obtaining bioactive compounds and fit into the concept of a circular economy and sustainable development.

## 2. Materials and Methods

### 2.1. Raw Material and Organization of Experiments

The flowchart of conducted experiments is shown in detail in Figure 1. Briefly, MAE was used for the optimization of pectin extraction from TPW, and the optimized process was compared to CSE regarding yields, pectin quality and the environmental footprint. TPW was obtained as a by-product after the preparation of tomato salsa from the Faculty of Agrobiotechnical Sciences Osijek, Croatia. Before extraction, TPW (seeds, skin, parts of the pericarp and placenta) was dried in a laboratory oven at 70 °C for 48 h. After drying, the pomace was milled in a laboratory mill, sieved through a 0.8 mm sieve and stored at 4 °C until extraction. The importance of TPW pretreatment on the yields and quality of the obtained pectin was investigated by applying: mechanical ball milling (1); different drying conditions (2) and defatting (3) prior to MAE. For mechanical ball milling, a vibrating MM 500 control micromill (Retsch, Haan, Germany) was used. Briefly, 8 g of TPW was ground together with 12 large balls (10 mm) and 24 small balls (5 mm) at the frequency of 30 Hz for 15 min. The particle size, expressed as the volume distribution, of the samples milled in the vibrational micromill and laboratory mill was determined using laser diffraction (Mastersizer 3000, Malvern Instruments, Worcestershire, UK). The instrument is equipped with a Hydro SV dispersion unit with a magnetic stirrer. A small amount of the sample was dispersed in 10 mL of distilled water and added to the Hydro SV unit until 10–20% obscuration. The results were presented as an average of five replicates. For the comparison of different drying procedures, the samples were dried in a laboratory dryer (Inko, Zagreb, Croatia) at 70 °C for 48 h (1); in a laboratory dryer at 50 °C (for 72 h) (2); in a 21GV vacuum dryer at 35 °C (Fratelli Galli, Fizzonasco, Italy) for 72 h (3); and by lyophilization for 24 h (Epsilon 2–4 LSCplus, Martin Christ Gefriertrocknungsanlagen GmbH, Osterode am Harz, Germany), with rapid freezing conducted at −50 °C and primary drying at −15 °C (4). The defatting of samples was conducted using the continuous 13 h hour extraction method in a Soxhlet apparatus, with petroleum ether as the solvent.

### 2.2. Experimental Design

An experimental design was employed for MAE experiments, and the response surface methodology was used to model the response variables, with the aim of optimizing their outcomes. Stat-Ease-360 22.0.8. software (Stat-Ease, Minneapolis, MN, USA) was utilized for designing experimental runs and data processing. For CSE, previously established optimal conditions were adopted from the existing literature [18].

The central composite design model with 6 predefined central points was applied to optimize pectin extraction by microwave heating. Three independent variables were tested during the microwave extraction experiments: extraction time, pH and microwave power (Table 1). Each independent variable was tested at five levels: –α, –1, 0, 1 and +α, as summarized in Table 1. Preliminary experiments were conducted multiple times to determine the appropriate range for each variable. The results from 19 experimental conditions served as the input data.

### 2.3. Pectin Extraction

#### 2.3.1. Conventional Solvent Extraction of Pectin

The extraction of pectin from TPW was conducted as described by Casas-Orozco and co-workers with some modifications [19]. Briefly, 15 g of tomato pomace was extracted with 300 mL of 1% citric acid monohydrate (pH 1.5) for 2 h at a temperature of 85 °C in a water bath. The reaction mixture was filtered while hot through a cotton gauze and FN3 filter paper on a sinter funnel (grade 2) under vacuum, followed by another filtration with FN3 filter paper and the addition of 600 mL of 96% ethanol to the filtrate. The filtrate/ethanol solution was placed in the refrigerator for 24 h to allow for complete pectin precipitation. Pectin was collected through simple filtration, dried at 45 °C until constant weight, ground in a mortar and stored at 4 °C.

#### 2.3.2. Microwave-Assisted Extraction of Pectin

MAE was performed using the Ethos Easy microwave system (Milestone, Sorisole, Italy) with adjustable microwave power up to 1800 W, operating at 2.45 GHz. Each extraction started with a 10-s phase to reach the set microwave power, followed by a constant power phase for the designated extraction time, and ended with a 10-min cooling period. The stirring speed was set to 10%. The extractions were carried out following the experimental design outlined in Table 2. The temperature change was monitored during the process. A total of 1 g of TPW was mixed in the vessel with 20 mL of 1% citric acid monohydrate (with the pH adjusted to 0.32, 1, 2, 3 or 3.68 using either 6M hydrochloric acid or 6M sodium hydroxide). A 1:20 sample-to-solvent ratio (SSR) was chosen based on extensive literature research pointing out that this is the lowest possible SSR that can be applied for successful pectin extraction from TPW. It is important to point out that MAEis significantly lower compared to SSRs applied for CSE of TPW pectin ranging from 1:180 to 1:50 [8].

Vessels were then closed and placed in the microwave rotor plate. Afterwards, the extract was passed through a cotton gauze directly into a sinter funnel (grade 2) with FN3 filter paper, and the extract was vacuum filtered, followed by a second filtration with filter paper. To precipitate pectin, an amount of 96% ethanol equal to twice the volume of the extraction solvent was added to the filtrate, and the mixture was refrigerated at 4 °C. After 24 h, pectin was collected via filtration, dried at 45 °C until constant weight (approximately 36 h in the 0.5 cm thick layer), ground in a mortar and stored at 4 °C.

### 2.4. Pectin Characterization

#### 2.4.1. Pectin Yield

Pectin yields from both CSE and MAE were calculated using Equation (1):(1)$$\mathrm{Pectin}\;\mathrm{yield}\;\left(\%\right)=\frac{w\;\left(\mathrm{pectin}\right)}{w\;\left(\mathrm{TPW}\right)}\times 100$$
where w (pectin) is dry weight of pectin and w (TPW) is the dry weight of tomato pomace waste.

#### 2.4.2. Chemical Characterization of Pectin

Obtained pectin fractions were characterized in terms of anhydrouronic acid content (AUA), degree of esterification (DE), methoxyl content (MC) and equivalent mass (EM) according to standard analytical methods [20,21,22].

#### 2.4.3. Fourier Transform Infrared Spectroscopy

Fourier transform infrared (FTIR) spectroscopy was applied to identify the characteristic functional groups (COOR, ester, COO- and carboxylate) of pectin obtained under optimized extraction conditions. The pectin was analyzed using an FTIR spectrophotometer (Spectrum Two, Perkin Elmer, Waltham, MA, USA) over a spectral range of 500–4000 cm^−1^ with 16 scans and a spectral resolution of 4 cm^−1^.

### 2.5. Modeling of Environmental Impact Using LCA

The evaluation of the environmental impact of tomato pomace treatment conducted under two extraction methods (conventional and microwave) was performed using a partial LCA approach from “gate-to-gate”, i.e., in lab-scale conditions. The LCA comprised of mapping the process, setting the lab-scale scope and boundaries, defining the inventory, calculating the environmental footprint and analyzing the results of the two treatments [23]. Two functional units (FUs) were used as output references: (i) one treatment of tomato pomace as extraction FU and (ii) pectin mass [g] as nutritional FU. The system boundaries of this LCA are presented in Figure 2.

The inventory analysis included the use of natural resources (water and electric energy) and the use of chemicals, namely ethanol and 1% citric acid monohydrate (as solvents). Regarding the energy needed for the drying process of tomato pomace and pectin that was performed in the same equipment at the same time, mass allocation was used to define the energy consumption. The outputs covered liquid organic waste (mainly mixed with citric acid and ethanol) and solid organic waste (dry tomato waste). The calculation of the environmental impacts was performed using data from LCA databases available in openLCA (GreenDelta GmbH, Berlin, Germany) by employing the environmental footprint (EF) as the selected life cycle impact assessment method. The environmental impact of tomato pomace (as the input) was not included in this study to avoid the potential influence of this impact on the extraction process, as the production of similar food differs depending on the quality of data in both primary production and processing [24]. Environmental footprints calculated in this study covered the following: global warming potential; acidification, the use of non-renewable (fossil) resources and freshwater eutrophication, as outlined by Gaffey et al. [25]; and ozone depletion, cancerogenic and non-cancerogenic human toxicity and ionizing radiation.

### 2.6. Data Analysis

The analyses in this study were performed in duplicates for pectin extraction and in quadruplicates for pectin characterization. Results were presented as mean values ± standard deviations. Statistical analyses were conducted using the Microsoft Excel (Microsoft 365) and GraphPad Prism 10.4.1 software. The differences between groups were tested using Student’s *t*-test or one-way ANOVA followed by Tukey’s post hoc multiple comparison test, depending on the number of groups. A *p*-value of <0.05 was considered statistically significant.

## 3. Results and Discussion

### 3.1. Optimization of Pectin Extraction

#### 3.1.1. Fitting the Model

The response surface methodology with a central composite design was utilized to model the influence of selected independent variables on the pectin yield (%), AUA (%) and DE (%) (Table 2).

**Table 2 foods-14-01516-t002:** Central composite design matrix and observed responses.

	Independent Variables			Experimental Values		
	Variable 1: Time (min)	Variable 2: Power (W)	Variable 3: pH	Yield (%)	AUA (%)	DE (%)
1	12	1200	1	0.00 ± 0.00	0.00 ± 0.00	0.00 ± 0.00
2	12	600	3	6.09 ± 0.09	24.56 ± 1.68	53.08 ± 2.25
3	7	900	2	7.09 ± 0.08	25.73 ± 1.55	45.72 ± 1.50
4	7	395	2	4.25 ± 0.04	39.49 ± 2.85	34.62 ± 1.37
5	7	900	2	7.04 ± 0.46	28.95 ± 1.12	39.67 ± 1.39
6	7	900	2	5.83 ± 0.12	20.87 ± 0.26	31.46 ± 0.96
7	2	1200	1	4.93 ± 0.06	32.69 ± 1.24	25.26 ± 0.75
8	2	600	1	3.29 ± 0.12	41.08 ± 9.23	31.64 ± 1.49
9	0	900	2	0.00 ± 0.00	0.00 ± 0.00	0.00 ± 0.00
10	7	900	2	6.83 ± 0.37	25.90 ± 0.49	36.06 ± 0.87
11	12	1200	3	4.03 ± 0.15	26.56 ± 1.21	48.52 ± 2.50
12	2	600	3	3.96 ± 0.19	39.63 ± 1.34	45.48 ± 0.22
13	7	900	3.68	8.27 ± 0.43	28.98 ± 1.26	47.49 ± 1.11
14	12	600	1	10.08 ± 0.20	38.02 ± 1.00	35.13 ± 0.62
15	7	900	2	6.64 ± 0.35	29.70 ± 2.01	42.63 ± 0.61
16	7	900	0.32	6.22 ± 0.49	49.92 ± 4.85	14.37 ± 0.24
17	7	1405	2	6.48 ± 0.19	35.10 ± 1.29	44.97 ± 1.12
18	7	900	2	7.04 ± 0.34	25.54 ± 1.61	46.14 ± 1.38
19	15.41	900	2	4.58 ± 0.03	30.93 ± 0.22	48.92 ± 1.96
20	2	1200	3	4.87 ± 0.13	25.26 ± 0.55	51.56 ± 0.52

AUA—anhydrouronic acid content; DE—degree of esterification.

The quadratic model (*p* = 0.052) best described the influence of time, microwave power and pH on pectin yield. R^2^ = 0.7285 indicates satisfactory explanation of the variance of the dependent variable. The model successfully recognized interactions between independent variables that had a significant influence on pectin yield. The quadratic term A^2^ (time^2^, *p* = 0.071) indicates that the influence of time follows a reverse U shape curve, meaning that the yield increases up to a certain point, after which a further increase in extraction time starts to have a negative impact on the yield.

DE was optimally described by the linear model (*p* = 0.0140, R^2^ = 0.4754), with pH (*p* = 0.0026) having the strongest influence on the observed outcome. A decrease in pH led to lower DE, which was expected since ester bonds in a highly acidic environment are susceptible to hydrolysis [26]. Our model could not describe the influence of selected independent variables on AUA. It can be presumed that other variables, presumably pectin source, purity, extraction temperature and extraction solvent, are stronger determinants of AUA compared to the independent variables tested in this work [22,27,28].

Model statistics and coefficient estimates can be found in Appendix A (Appendix A). The main disadvantage of our model is that it did not account for the temperature as one of the possible variables to influence the analyzed outcomes because of the construction of the microwave extraction unit that is usually (as in this case) not equipped with the cooling system. Previous research indicates that temperature is a strong determinant of pectin extraction yield [9,29]. Even though most of the research reports the ability to control both the temperature and the power (as independent variables) when performing microwave extractions, there is, in fact, a lot of variation. For example, when trying to control both variables, the temperature will reach a specified value, and microwave power will remain lower than specified to prevent the overheating of the system. Even if the specified microwave power could be reached and maintained for some time, it would start to decrease to avoid the unwanted increase in temperature. When trying to achieve the specified temperature and power, it is necessary to have long periods of the pre-extraction phase to allow the system to stabilize; however, during this phase, the sample is already impacted by heat and the microwave power, which are not controlled. These technical difficulties are rarely (if never) addressed in similar works. In our work, when specifying only the microwave power, the value was reached within the first 10 s of the run, resulting in a shorter adjustment phase and a more reliable control of the conditions. Therefore, in our investigation, the emphasis was put on specifying the value of microwave power and monitoring the temperature of the reaction. The highest yield was obtained in the temperature interval between 90 °C and 120 °C. Under lower temperatures, the reaction conditions were not sufficient for optimal pectin extraction, while at the higher temperature, pectin is susceptible to degradation.

To the best of our knowledge, this is the first time the response surface optimization method was used to model the MAE of pectin from tomato pomace. Previous research focused on the optimization of extraction from different materials. For example, Spinei and Oroian [30] used the Box–Behnken RSM design to optimize the MAE of pectin from grape pomace under lower power (280–560 W) and shorter reaction times (60–120 s), while pH was set to values as in our work (1–3). Dependent variables (yield, DE, AUA and molecular weight) were explained by quadratic models, which, in our case, was shown only for the yield variable. However, they did not report the temperature of the reaction; they only stated that it increases with the higher power of the microwave radiation.

Dao et al. [31] investigated the optimal MAE conditions of pectin from dragon fruit and passion fruit peels. The Box–Behnken design was used to model the optimization of pectin yield. They reported that the quadratic model best fits the data, with independent variables being microwave power, pH, extraction time and liquid–solid ratio. The equipment used in the research enabled the extraction to be performed at a constant temperature of 80 °C, with different microwave power levels applied. Since both the power and temperature presumably strongly influence the yield, the ability to control both variables resulted in a model that can well describe the extraction process.

#### 3.1.2. Multiple Response Optimization and Response Surface Analysis

In multiple response optimization, priority was given to maximizing the yield and AUA, which were designated with the highest importance during the optimization process. DE was given medium importance, and the acceptable interval was set between 30 and 70%. Multiple response optimization is shown in Figure 3.

Response surface graphs for the yield are presented in Figure 4. They demonstrate the influence of two independent variables while keeping the third independent variable constant. The combination of the extreme values of time and power (pH = 1) has a negative impact on the yield, which is also recognized in the model statistics (time × power, *p* = 0.0144). Our results indicate that lowering the power and increasing reaction time is a better choice for maximizing the yield than a shorter reaction time under high microwave power. Under low power conditions (600 W), pH had a limited influence on the pectin yield in the context of extraction time. Figure 4C shows that extreme power and pH (time = 11.66 min) greatly reduced the pectin yield. The optimal conditions were 11.66 min, 600 W and a pH of 1. Spinei and Oroian reported the following optimal conditions: 2 min, 560 W and a pH of 1.8 [30]. These conditions are considerably milder compared to those in our research, which is probably due to the combination of different levels of tested factors in the optimization as well as the difference in the extraction material. Both high and low power of microwave radiation have been reported in the literature as optimal conditions for pectin extraction. Studies performed by Swamy and Muthukumarappan, who used banana peels [32], and Rahmani et Rahmani et al., who used sweet lemon peels as the pectin source [33], describe using high microwave power for maximal pectin yield. Conversely, Sarah Et Al. reported low microwave power as optimal for MAE pectin extraction [34]. The choice is therefore to apply high power and a shorter time or the opposite, that is, low power and a longer extraction time. The optimal power level in our work was comparable to the high-power condition in the mentioned research, although it was set to a low factor level in the design. Generally, the conditions reported here were relatively strong compared to the other studies. Finally, it is important to mention that the investigation of optimal conditions can be influenced by the limitations of the equipment used.

The predicted and observed values for the conformation run are shown in Table 3. The impact of the tested variables on the yield was accurately predicted, which was the primary goal of the extraction optimization process. As previously mentioned, the influence on AUA could not be described and predicted by our model, which is further evidenced from the data in the table. DE is very slightly above the high confidence interval; however, in the process of optimization, the acceptable interval of DE was set between 30 and 70%, and the observed mean aligns with these conditions.

### 3.2. Comparison of TPW Pectin Obtained by CSE and MAE

Both the yields and properties of pectin are significantly influenced by its origin (the raw material from which pectin is derived), as well as the extraction conditions/type. A recent review by Radić and co-authors showed that pectin yields from tomato pomace vary from 4.3 to 35.7%, depending on the type and conditions of extraction. Pectin properties also varied significantly: AUA varied from 125 to 572 g/kg, DM from 21 to 88%, DE from 53 to 89% and molecular weight from 31 to 510 kDa [8].

In this work, we compared the yields and properties of pectin obtained via CSE or novel MAE, and the parameters that were investigated included yield, the EM of pectin, MC and DE, which were determined using chemical methods. As described previously, the equivalent masses of pectin and methoxyl content were used for the calculation of AUA. The obtained results are presented in Figure 5.

Yields obtained using CSE and optimized MAE differed significantly (Figure 5A) and indicate clearly the advantage of utilizing MAE for obtaining pectin. Yields obtained using optimized MAE are comparable with the results by Lasunon and co-workers [15] and Sengar and co-workers [16], who also investigated the applicability of MAE for the extraction of pectin from TPW; pectin yields obtained using short (<10 min) MAE treatment ranged between 9.4 and 15.1%.

The EM of MAE-obtained pectin was significantly lower compared to CSE pectin (778.5 g/mol vs. 940 g/mol), indicating different functional properties of TPW pectin obtained using different extraction methods. The equivalent mass provides insights into how effectively pectin can form gels; a lower equivalent mass indicates a greater number of reactive sites available for divalent cations (such as calcium), which increases the strength and stability of the gel [35].

The impact of the type of extraction on the quality and functional characteristics of pectin is less investigated; therefore, the comparison of data obtained in this research and values reported in the literature by other authors was not possible. As shown in Figure 5, pectin obtained via MAE contained a higher content of AUA, indicating higher purity (50.0% vs. 32.6%). Also, MAE pectin was organoleptically different from CSE pectin; its consistency was more liquid, probably due to lower DE (46.8% in CSE pectin vs. 43.7% in MAE pectin). Despite these differences, both types of pectin can be classified as low-methoxyl pectin, which means that TWP pectin generally forms softer gels in the presence of calcium ions without the need for sugar, making it suitable for low-sugar or sugar-free products. Also, DE affects the physiological effects of pectin; low-methoxyl pectin is often associated with health benefits such as cholesterol reduction (due to its ability to act as soluble dietary fiber and prebiotic) [35]. However, it is important to emphasize that during the optimization of MAE, we proved that by increasing the pH, it is possible to obtain high-methoxyl pectin but at the expense of a lower pectin yield (Table 2).

The properties of pectin obtained using MAE and CSE were also compared based on their respective FTIR spectra (Figure 6) that contained characteristic peaks corresponding to the vibrations of characteristic functional groups/chemical bonds that confirm the structure of pectin.

Characteristic peaks at 3315 cm^−1^ (CSE) and 3284 cm^−1^ (MAE) are stretching vibrations of the O-H group that indicate a high level of hydration and the presence of free hydroxyl groups of pectin. A stronger signal of MAE, compared to CSE-derived pectin, indicates a greater presence of free hydroxyl groups and/or a higher level of hydration compared to pectin, which is consistent with previously presented data obtained through chemical analysis of pectin, which show that MAE pectin has a lower degree of esterification compared to CSE pectin (Figure 5B). Peaks at 2926 cm^−1^ (CSE) and 2924/2854/1450 cm^−1^ (MAE) are C-H stretching vibrations of aliphatic hydrocarbons, indicating the presence of methyl and methylene groups. Peaks at 1739 cm^−1^ and 1742 cm^−1^ are stretching vibrations of the C=O bond from ester groups, indicating the presence of esterified carboxylic acid [36]. A slightly stronger signal of MAE pectin at 1739 cm^−1^ indicates a more pronounced presence of C=O bonds, which is consistent with the data showing a slightly higher proportion of AUA in MAE pectin, as shown via chemical characterization (Figure 5B). Peaks at 1642 cm^−1^ (CSE) and 1627 cm^−1^ (MAE) are stretching vibrations of C=C bonds within the polysaccharide structure or from aromatic compounds. The peak at 1530 cm^−1^ is the stretching vibrations of N-H bonds, indicating the presence of amino or amide groups from proteins or peptides in the sample. Peaks visible at 1230/1147/1076 cm^−1^ (CSE) and 1077/1150 cm^−1^ (MAE) are characteristic for carbohydrates, indicating the vibrations of the C-O (glycosidic) bond [36].

The obtained results are consistent with the observations of other authors who investigated the compatibility of MAE for the extraction of pectin from other types of raw materials. Recent literature data focusing on other pectin sources confirm that applying MAE instead of CSE significantly impacts not only yields but also the chemical/functional characteristics of pectin. Mayid and co-workers [37] showed that applying MAE increases yields and AUA and has no effect on the DE of watermelon rind pectin. Similarly, in comparison to CSE, higher pectin yields and higher AUA were obtained when MAE was applied for obtaining pectin from pomelo peels and grapefruit, while DE remained unaffected [38,39]. Košťálová and co-workers [40] optimized the extraction of pectin from pumpkin seeds and also showed that CSE results in lower pectin yields and a higher content of non-pectin polysaccharides (i.e., lower AUA). Additionally, CSE-extracted pectin had a higher equivalent mass, which is consistent with our results.

Observed differences in the quality and functional characteristics of pectin obtained via CSE and MAE can be attributed to the differences in the pH, duration and temperature of the extraction process. Higher yields obtained via MAE might in part be due to lower pH (1 vs. 1.5), which is consistent with the results obtained by other authors [15,16,17,29]. The lower equivalent mass of MAE pectin (729.4 g/mol vs. 940 g/mol) is probably also due to the lower pH used for MAE, which caused the depolymerization of pectin chains and affected pectin’s gelling properties [32]. Even though MAE was significantly shorter compared to CSE (11.66 min vs. 120 min), the extraction efficiency in MAE was enhanced and resulted in an increased yield despite shorter extraction times. This is probably because the utilization of microwaves enables the solvent to be heated rapidly, which favors pectin extraction. However, the observed differences in pectin yields and properties are at least partially attributed to different temperature extraction conditions. CSE was conducted at a constant temperature of 85 °C.

As mentioned previously, in the case of MAE optimization, microwave power and the duration of extraction were predetermined, while the temperature change could not be manipulated, only monitored (Figure 7).

Figure 7A shows the temperature change in different runs. (Extraction conditions for every run are presented in Table 2.) Maximal temperatures achieved in different runs ranged from 64 °C (run 8 and run 12, performed at 600 W for 2 min) up to 170 °C (run 1 and run 11, performed at 1200 W for 12 min) and were the function of applied power and the duration of the extraction process, as explained previously. Even though response surface analysis showed no significant correlation between pH and the total pectin yield, data shown on Figure 7B indicate that at lower pH, comparable yields are achieved at lower peak temperatures. This is evident when comparing yields of runs 8 and 12 (3.29% vs. 3.96%), runs 2 and 14 (6.09% vs. 10.07%) or runs 1 and 11 (0% vs. 4.03%). The increased temperature of the reaction mixture (as a function of the applied microwave power and the duration of extraction) improved yields up to a certain peak temperature, while a further increase caused a reduction in yields, probably due to pectin degradation. This is consistent with the results of the response surface analysis that showed that the dependence of pectin yields on the power and duration of extraction can be described by a quadratic function (Figure 4, Eq 1). This threshold temperature ranged from 114 °C to 121.5 °C, depending on the pH of the reaction mixture.

### 3.3. Impact of TPW Pretreatment on the Yield and Characteristics of Pectin Obtained via Optimized MAE

Different pretreatment technologies for biomass (biological, mechanical and chemical) can significantly influence the yields and quality of the desired product, as well as the economic and environmental sustainability of the extraction process, and their selection depends on numerous factors: foreseen utilization of the main biomass components, efficiency in separation, environmental concerns, compatibility with subsequent processes, energy requirements and costs [41,42]. The majority of the available literature data are focused on the available pretreatment techniques for obtaining fermentable sugars from lignocellulosic biomass. However, data on the impact of different pretreatments of TPW on the yields of bioactive compounds, including pectin, are very scarce. As mentioned in the Introduction Section, HPE treatment of TPW prior to pectin extraction increased the yields and influenced the molecular weight of the obtained pectin [17]. Therefore, this study investigated if modifying/optimizing common biomass pretreatment procedures (defatting, drying and milling) can significantly influence the yields and the quality of the obtained pectin. The impact of defatting and pretreatment in a vibrational micromill on the yields, AUA and DE of TPW-derived pectin is presented in Figure 8, and the impact of different drying conditions (50 °C, 70 °C, lyophilization and vacuum drying) is presented in Table 4.

The impact of defatting the biomass on the yields and characteristics of TPW pectin was investigated because usually, the extraction of dietary fibers from biomass involves this step. However, its necessity is dependent on the amount of fat in the raw material. Omitting this step from the process might significantly increase both the economic and environmental sustainability of the extraction process by reducing process costs, increasing the overall value of the remaining processed biomass for other purposes and reducing the overall environmental footprint due to lower resource utilization.

Our results show that the defatting of TPW prior to extraction increases pectin yields by 10%, slightly increases its purity and has no significant effect on DE (Figure 8B).

The particle size distribution of TPW milled in a laboratory mill and used for CSE and MAE is presented in Appendix A. In order to investigate the importance of particle size on the pectin yield and quality, milled TPW was additionally treated in a vibrational micromill, which resulted in a significant decrease in the average particle size (Appendix A). This had a statistically significant effect on pectin yield and purity (Figure 8A) but had no effect on DE. This is consistent with the conclusions of the recent publication by De Laet and co-workers [43] who also showed that particle size reduction results in an increase in the pectin extraction yield. Similarly, the extent of milling influences the extraction efficiency of dietary fibers like arabinoxylans and cellulose [44,45].

Drying is an energy-intensive pretreatment process that impacts biomass sanitary safety and quality and therefore has a significant impact on the economy and sustainability of biomass utilization [46]. Even though there are no data about the impact of the drying method on the yield and quality of TPW data, a work by Siriwattananon and Maneerate [47] shows that drying can significantly influence the content of dietary fiber in some fruits and vegetables. This was not the case in our investigation; pectin yields from conventionally dried, vacuum-dried and freeze-dried TPW varied from 13.9 ± 0.21 to 15.4 ± 0.07%, but observed changes were not statistically significant. Pectin quality parameters (EM, AUA and DE) also varied, but even when observed changes were statistically significant, they were minor (less than 7%) (Table 4). This is partially due to the fact that TPW was obtained from tomatoes thermically processed at 100 °C; therefore, temperature differences in subsequent processing were not significant. Also, pectin is not a thermolabile substance and is extracted at high temperatures. Therefore, when optimizing the drying of TPW, focus should be set on reducing energy consumption and/or optimizing the content of other valuable compounds such as carotenoids or polyphenols.

### 3.4. Environmental Impact of Tomato Pomace Treatment

The reutilization of biomass, such as TPW, for obtaining bioactive compounds is in line with the concept of a circular economy, particularly when contemporary green processing methods are applied. However, when developing and validating a novel process, it is crucial to conduct the objective and comprehensive estimation of their ecological acceptability, based on quantifiable data instead of relying on subjective or partial assessments. The verification of different green extraction processes when upcycling and valorizing food waste and by-products is needed, and the most comprehensive approach for such estimations is an LCA [48]. When an LCA is employed in lab-scale conditions, it is important to understand its technology readiness level, thus enabling a better understanding of the overall environmental impact [49]. This methodology identifies the hotspots that contribute the most to the overall environmental impact, allowing for targeted improvements and enabling the comparison of different processes, based on their environmental performance, thereby allowing for informed decision-making on the most sustainable options.

Although MAE may be generally considered as a “green extraction method” that shortens extraction times and utilizes lower amounts of solvent, in this work, we used an LCA to objectively compare the environmental footprint of CSE and optimized MAE. Table 5 depicts the results of eight selected environmental footprints of the two extraction methods. Based on the inventory, the results show that one conventional extraction treatment has higher environmental impacts compared to one microwave extraction treatment for all the indicators, more than twice as much. Bearing in mind that conventional extraction needs more tomato pomace than microwave extraction (35 g compared to 15 g), the pectin yield for the first extraction is much lower compared to the second (4.81% for 1.68 g vs. 9.43% for 1.42 g). This is also visible in the environmental footprint results regarding the second functional unit (the mass of the extracted pectin [g]), where the values of the environmental impacts are higher (almost double) than those of conventional extraction.

To our knowledge, there is only one available similar research, focused on extracting lycopene from TPW, that utilized LCA and techno-economic analysis to objectively assess and compare the economic and environmental implications of newly developed green extraction processes (but not MAE) [50]. This study revealed that various “green” extraction techniques differ significantly regarding costs and the environmental footprint and showed that ultrasound surfactant-assisted extraction surpasses other methods in terms of economic viability and the lowest environmental impact.

## 4. Conclusions

The findings of this study demonstrate that MAE is a highly efficient and environmentally sustainable method for recovering pectin from TPW. Optimized MAE conditions (11.66 min, 600 W and a pH of 1) yielded significantly higher amounts of pectin compared to CSE while also reducing the ecological footprint of the process. The LCA confirmed that MAE has a lower environmental impact across all indicators, making it a promising green technology for industrial applications. Furthermore, pretreatment methods such as mechanical ball milling and defatting were found to have negligible effects on pectin yields and properties, suggesting that these energy-intensive steps can be omitted to further enhance process sustainability. The chemical characterization of the extracted pectin revealed satisfactory functional properties, supporting its potential use in food and pharmaceutical industries. This research contributes to the valorization of food industry by-products, aligning with the principles of a circular economy and sustainable development. By optimizing MAE for pectin extraction, this study provides a scalable solution for utilizing TPW as a valuable resource, paving the way for the broader industrial adoption of green extraction technologies.

## Figures and Tables

**Figure 1 foods-14-01516-f001:**
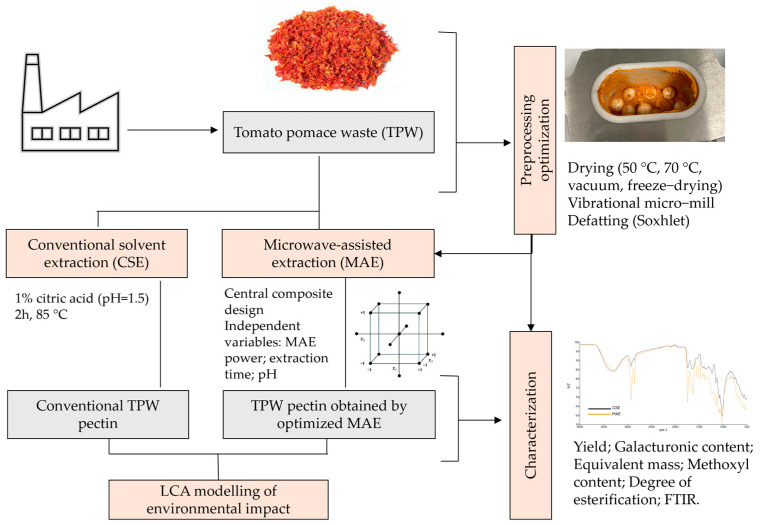
Flowchart of the conducted experiments.

**Figure 2 foods-14-01516-f002:**
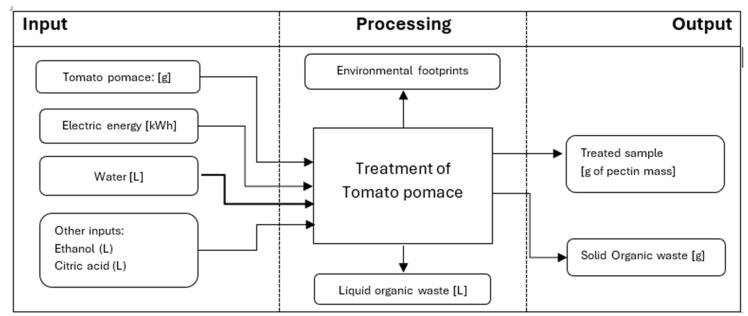
Generic model of tomato pomace extraction.

**Figure 3 foods-14-01516-f003:**
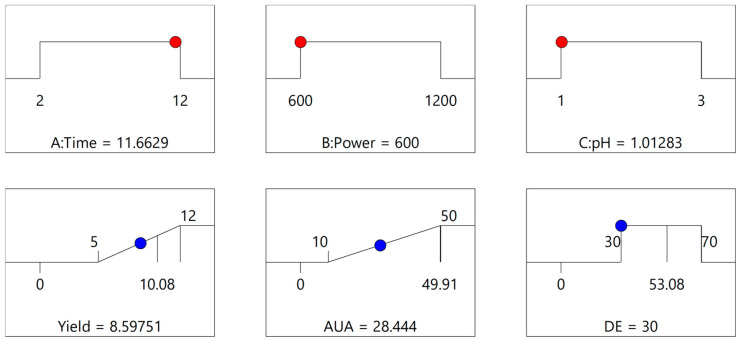
Multiple response optimization. The red colored dots represent optimal values of independent variables and blue colored dots represent predicted values of response variables.

**Figure 4 foods-14-01516-f004:**
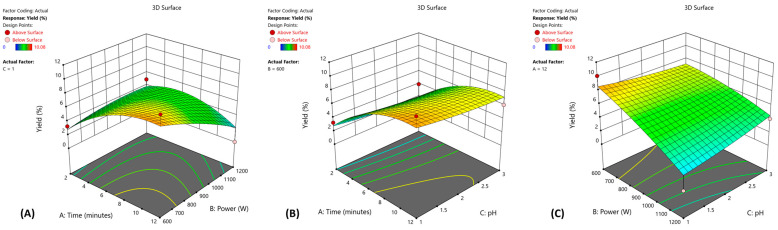
Response surface plots showing the effect of microwave power and time (**A**); pH and time (**B**); pH and microwave power (**C**) on pectin yield.

**Figure 5 foods-14-01516-f005:**
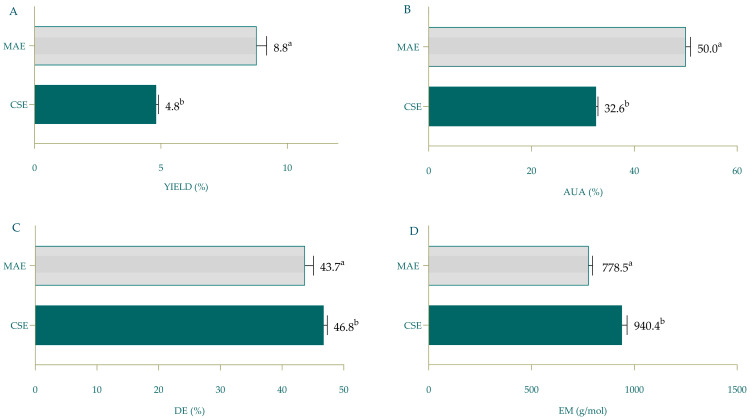
Comparison of yield (**A**), AUA (**B**), DE (**C**) and EM (**D**) of pectin obtained using optimized MAE (pH 1, 600 W, 11.66 min) and CSE. Columns marked with different letters belong to different statistical groups (*p* < 0.05). MAE—microwave-assisted extraction; CSE—conventional solvent extraction; AUA—anhydrouronic acid; DE—degree of esterification; EM—equivalent mass.

**Figure 6 foods-14-01516-f006:**
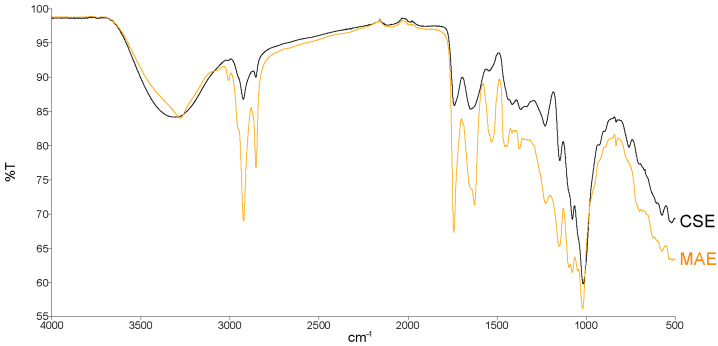
Comparison of FTIR spectra of tomato pectin obtained by MAE and CSE. MAE—microwave-assisted extraction; CSE—conventional solvent extraction; T—transmission.

**Figure 7 foods-14-01516-f007:**
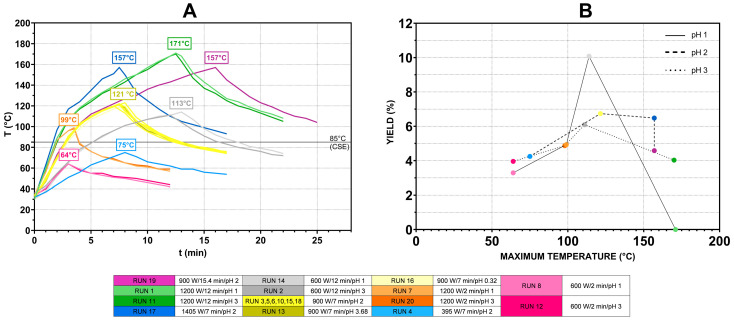
Temperature changes during MAE (**A**) and their impact on pectin yields at different pH values (**B**).

**Figure 8 foods-14-01516-f008:**
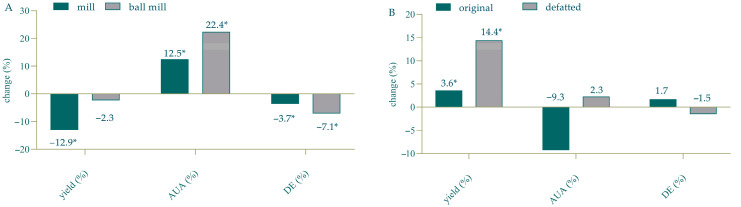
Impact of pretreatment in vibrational micromill (**A**) and defatting (**B**) on the yield, AUA and DE of TPW derived pectin. AUA—anhydrouronic acid content; DE—degree of esterification; * observed difference is statistically significant (*p* < 0.05).

**Table 1 foods-14-01516-t001:** Independent variables and their levels for the central composite design of experiments.

Independent Variables		Levels				
**Uncoded**	**Coded**	**−α**	**−1**	**0**	**+1**	**+α**
**Microwave power (W)**	X1	395	600	900	1200	1405
**Extraction time (min)**	X2	0	2	7	12	15.4
**pH**	X3	0.32	1	2	3	3.68

**Table 3 foods-14-01516-t003:** Predicted and observed values of dependent variables under optimal extraction conditions.

Analysis	Predicted Mean	SE Mean	95% CI Low	95% CI High	Observed Mean
**Yield**	8.59 ± 1.75	1.37	5.53	11.66	8.79 ± 0.39
**AUA**	28.44 ± 12.04	2.69	22.8	34.08	50.03 ± 0.91
**DE**	30.00 ± 12.32	6.28	16.67	43.32	43.71 ± 1.37

AUA—anhydrouronic acid content; DE—degree of esterification.

**Table 4 foods-14-01516-t004:** Impact of different drying conditions on yields and characteristics of TPW pectin obtained by optimized MAE.

	Conventional Drying		Vacuum Drying	Freeze Drying
	70 °C	50 °C		
Yield (%)	13.9 ± 0.21 ^a^	15.4 ± 0.07 ^a^	14.8 ± 0.00 ^a^	15.2 ± 0.90 ^a^
EM (g/mol)	782.9 ± 28.1 ^a^	706.2 ± 53.4 ^b^	805.9 ± 3.36 ^a^	799.0 ± 9.07 ^a^
MC (%)	4.43 ± 0.30 ^ab^	3.49 ± 0.46 ^a^	4.10 ± 0.49 ^ab^	4.77 ± 0.79 ^b^
AUA (%)	47.6 ± 2.25 ^a^	44.9 ± 4.31 ^a^	45.1 ± 2.79 ^a^	49.1 ± 4.21 ^a^
DE (%)	41.2 ± 0.95 ^a^	38.5 ± 0.71 ^b^	41.4 ± 1.29 ^a^	38.3 ± 2.04 ^b^

EM—equivalent mass; MC-methoxyl content; AUA—anhydrouronic acid content; DE—degree of esterification. Values in the same row marked with same letters belong to the same statystical group (*p* > 0.05). Results are expressed on dry matter.

**Table 5 foods-14-01516-t005:** Estimated environmental impacts of the two extraction methods expressed in two functional units.

		CSE	MAE	Index	CSE	MAE	Index
		FU1–One Treatment			FU2–Pectin [g]		
**Acidification**	mol H+ eq	0.29	0.13	2.14	0.17	0.09	1.81
**Climate change**	kg CO_2_e	38.97	18.15	2.15	23.20	12.78	1.81
**Freshwater eutrophication**	kg P eq	2.69 × 10^−3^	1.16 × 10^−3^	2.33	1.60 × 10^−3^	8.14 × 10^−4^	1.97
**Human toxicity-cancerogenic**	CTUh	5.81 × 10^−8^	2.62 × 10^−8^	2.22	3.46 × 10^−8^	1.84 × 10^−8^	1.88
**Human toxicity-non cancerogenic**	CTUh	1.28 × 10^−6^	5.91 × 10^−7^	2.17	7.63 × 10^−7^	4.16 × 10^−7^	1.83
**Ionizing radiation**	kBq U-235	3.00	1.40	2.14	1.79	0.99	1.81
**Ozone depletion**	kg CFC11 eq	6.55 × 10^−8^	2.81 × 10^−8^	2.33	3.90 × 10^−8^	1.98 × 10^−8^	1.97
**Use of fossil resource**	MJ	587.96	272.66	2.16	349.97	192.01	1.82

CSE—conventional solvent extraction; MAE—microwave-assisted extraction; FU1: functional unit is one treatment, FU2: functional unit is 1 g of extracted pectin.

## Data Availability

The original contributions presented in the study are included in the article/Appendix A, further inquiries can be directed to the corresponding author/s.

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
