# Peer review of "Microwave-Assisted Valorization of Tomato Pomace for Pectin Recovery: Improving Yields and Environmental Footprint"

_foods, 2025, doi:10.3390/foods14091516_

Round 1

Reviewer 1 Report

Comments and Suggestions for Authors

Pectin is a widely used additive in the food industry. The development of new, efficient and environmentally friendly pectin extraction methods is therefore has great practical relevance. Because of its high extraction yield, lower solvent demand and reduced process time requirement microwave assisted extraction can be a good alternative for conventional hot water extraction. Authors investigate the environmental impact of extraction methods using LCA. In my opinion, this provides interesting information not just for the science but also for the practice. The manuscript has a logic structure and contains interesting results but need revision to make it more complete and clear (see my comments).

Comments, suggestions:

Please discuss the amount (ratio) of waste material stream in tomato processing.

Please give clearly (in 2-3 sentences) the novelties of the study in the Introduction section as well.

In my opinion, the lower solvent requirement is one of the main advantages of the MAE method compared to conventional hot water extraction. Why did not you investigate the effect of the solid/solvent ratio?

Please give the details of the applied drying methods in methodology section.

Please improve the visibility of Figure 3 (mainly axis titles, units, labels)

Establishments in lines 390-469 need references (comparison etc.).

Author Response

RESPONSES TO REWIEVER 1
Pectin is a widely used additive in the food industry. The development of new, efficient and environmentally friendly pectin extraction methods is therefore has great practical relevance. Because of its high extraction yield, lower solvent demand and reduced process time requirement microwave assisted extraction can be a good alternative for conventional hot water extraction. Authors investigate the environmental impact of extraction methods using LCA. In my opinion, this provides interesting information not just for the science but also for the practice. The manuscript has a logic structure and contains interesting results but need revision to make it more complete and clear (see my comments).

 Comments, suggestions:

1.    Please discuss the amount (ratio) of waste material stream in tomato processing.
Answer. Thank You very much for the comment. Authors agree that emphasizing the wast avaialbility of tomato processing waste in Introduction section is important to point out the relevance of conducted investigation. The following sentence discussing the amount (ratio) of waste material stream in tomato processing has been added to Introduction (lines 66-69): “Tomato processing industries produce varying amounts of TPW, typically ranging from 2% to 10% of the total weight of processed tomato fruits. This waste primarily consists of peel, seeds, vascular tissues, and pulp residue. Based on these figures, the global tomato processing industry generates approximately 6.2 million to 9 million tons of TPW annually [8].”

2.    Please give clearly (in 2-3 sentences) the novelties of the study in the Introduction section as well.
Answer. Thank You very much for the comment. The novelty of the study has been described in the last section of the Introduction section (lines 127-138). However, authors agree that in the original version of the manuscript, the authors didn’t emphasize suficiently that none of the investigation tasks (goals) have been investigated before by other authors. Therefore this section has been rewritten to emphasize the novelty of the conducted study:
 “Considering all the above, the primary goal of this investigation was to optimize the MAE conditions to maximize the yields of TPW-pectin with satisfactory chemical characteristics. To our knowledge, this is the first attempt to develop the procedure using closed-vessel microwave extraction system that could result with higher pectin yields, shorter extraction times and reduced solvent utilization, compared to conventional solvent extraction, but also, compared to household microwave ovens used for this purpose by other authors [8]. Additionally, the paper investigated for the first time how different TPW pretreatments (drying, milling and defatting) can be optimized to improve pectin yields and decrease the environmental impact of the extraction process. In parallel, to analyze the potential of employing MAE as a “green” extraction solution, a life cycle assessment (LCA) has been used to compare the ecological footprint of MAE with CSE assessing the ecological acceptability of the proposed process. This is the first time, to our knowledge, that LCA was applied to realistically and objectivelly assess environmental advantages of MAE as “green extraction method” applied for pectin extraction. These goals contribute greatly to the attempts of reutilization of food industry waste for obtaining bioactive compounds and fit into the concept of circular economy and sustainable development”

3.    In my opinion, the lower solvent requirement is one of the main advantages of the MAE method compared to conventional hot water extraction. Why did not you investigate the effect of the solid/solvent ratio?
Answer. Thank You very much for this comment. Authors agree that this aspect hasn’t been mentioned nor explained in the paper. The choice of 1:20 (w:v) ratio was based on the extensive literature research that showed that for pectin extraction from tomato processing waste ratio solvent to tomato pomace ranges between 1:100 and 1:50. For extraction of pectin from tomato processing waste by appying UAE or MAE lower ratios can be applied leading to reduced solvent consumption. According to our literature research those ratios ranged from 1:50 to 1:20. (this has been reviewed by the same group of authors recently (https://doi.org/10.3390/su16219158).
Since 1:20 was the lowest sample to solvent ratio  we could find in the literature we decided to also use it in our research. We also conducted a preliminary experiment where we tested randomly chosen combinations of microwave power and duration for reaction mixtures containing 1:10 sample to solvent ratios – however after conducting extraction, we optained gelatinous samples (even carbonized when we applied higher powers and extraction times) that were impossible to process further to investigate characteristics of obtained pectin. Therefore we continued the experiments with the fixed sample to solvent ratio (1:20) that is consistent with the attempts of other investigators and also represents a significant reduction in solvent use compared to conventional solvent extraction. Authors agree that this should be shortly mentioned in the manuscript. Therefore the following sentence has been added to the Material and methods section (lines 232-236): 
“1:20 sample to solvent ratio (SSR) has been chosen based on the extensive literature research pointing out that this is the lowest possible SSR that can be applied for suc-cessful pectin extraction from TPW. It is important to point out that is significantly lower compared to SSRs applied for CSE of TPW pectin ranging from 1:180 to 1:50 [8].”

4.    Please give the details of the applied drying methods in methodology section.
Answer. Thank You very much for this remark. The details about different drying processes used for drying TPW have been added to the Material and Methods section (lines 166-173):
 “For the comparison of different drying procedures the samples were dried in a laboratory dryer (Inko, Zagreb) at 70 °C for 48 h (1);  in a laboratory dryer at 50 °C (for 72 h) (2); in a vacuum dryer at 35 °C (Fratelli Galli, 21GV, Italy) for 72 h (3); and by lyophilization for 24 hours (Martin Christ, Epsilon 2-4 LSCplus, Germany) with rapid freezing conducted at -50 °C and primary drying at -15 °C (4).”

5.    Please improve the visibility of Figure 3 (mainly axis titles, units, labels)
Answer. Thank You very much for this remark – it has been improved. We hav also corrected some minor mistakes in Figure 1 and Figure 4 ans uploaded new versions. 

6.    Establishments in lines 390-469 need references (comparison etc.).
Answer. Thank You very much for this remark. The authors have carefully revised the paragraph lines 390-469. In the original version of the manuscript we have provided all together 8 references in this section; one of those is the review focusing among other things on extraction methods applied for obtaining pectin from TPW. The existing research on this topic is scarce – therefore I am not sure if we can provide any more of relevant references for this part. However we included 2 more recent papers focusing on the connection between chemical characteristics of pectin and its functional properties  (doi.org/10.3390/foods11172683) as well as possibilities of application of spectroscopic data in pectin characterization (doi.org/10.3390/coatings12040546) that support our findings (statements). If You believe that particular aspevts of this part of the manuscript should be elaborated in more depth and substantiated with more literature data It would be helpful if You suggested more precisely whatyou have in mind and the authors will be happy to improve it. 

Reviewer 2 Report

Comments and Suggestions for Authors

  1. LINE 62, Suggest increasing the annual production of Tomato pomace waste (TPW) to better highlight the value of the research?
  2. Added analysis of data, eg.  this paper used quadratic polynomial equation to fit?
  3. LINE 287, Optimization of Pectin extraction.

Suggest adding the following three contents: 1)Response surface model; 2) Analysis of variance; 3)Multiple response optimization

  1. The discussion section lacks depth and requires additional references?

Author Response

Comments and Suggestions for Authors
1.    LINE 62, Suggest increasing the annual production of Tomato pomace waste (TPW) to better highlight the value of the research?
Answer. Thank You very much for this remark – it is in line with the remark of reviewer 1. In order to highlight the value of the research the following sentence has been included into the manuscript: 
(lines 66-69): “Tomato processing industries produce varying amounts of TPW, typically ranging from 2% to 10% of the total weight of processed tomato fruits. This waste primarily consists of peel, seeds, vascular tissues, and pulp residue. Based on these figures, the global tomato processing industry generates approximately 6.2 million to 9 million tons of TPW annually [8].”

2.    Added analysis of data, eg.  this paper used quadratic polynomial equation to fit? 
Answer. Thank You very much for this remark. If authors understood the remark correctly You asked to specifiy the models (functions) used to fit the data? This has been stated in lines 309-311 : ˝The quadratic model best described the influence of time, microwave power and pH on pectin yield. R2 = 0.7285 indicates satisfactory explanation of variance of dependent variable˝. and also in lines 344-345 : ˝Furthermore, DE was optimally described by linear model. pH (p = 0.0026) had the strongest influence on the observed outcome˝.

3.    LINE 287, Optimization of Pectin extraction
Answer. Thank You very much for this remark, it has been corrected

4.    Suggest adding the following three contents:
 1)Response surface model; 
2) Analysis of variance
 3)Multiple response optimization 
Answer. Thank You very much for this remark. The data You asked for have now  been inserted in the manuscript as supplementary material in Table S1, TableS2 and Figure S1. 

5.    The discussion section lacks depth and requires additional references?
Answer. Thank You very much for this remark. The authors have carefully revised the Ressults and Discussion section.  The authors would like to emphasize that the existing research on this topic is scarce – therefore I am not sure if we can provide any more of relevant references for this part. However we included 2 more recent papers focusing on the connection between chemical characteristics of pectin and its functional properties  (doi.org/10.3390/foods11172683) as well as possibilities of application of spectroscopic data in pectin characterization (doi.org/10.3390/coatings12040546) that support our findings (statements). If You believe that particular aspects of this part of the manuscript should be elaborated in more depth and substantiated with more literature data It would be helpful if You suggested more precisely what you have in mind and the authors will be happy to improve it.

Round 2

Reviewer 2 Report

Comments and Suggestions for Authors

The discussion section need deep improvement which the author didn't carry out. Suggest placing the attached chart in the main text. Divide 3.1 into two parts to improve its logic and facilitate readers' understanding.
eg. 3.1. Optimization of pectin extraction

3.1.1. Fitting the model

3.1.2. Analysis of response surface

Author Response

Comment 1.    The discussion section need deep improvement which the author didn't carry out. Suggest placing the attached chart in the main text. Divide 3.1 into two parts to improve its logic and facilitate readers' understanding.
eg. 3.1. Optimization of pectin extraction
3.1.1. Fitting the model
3.1.2. Analysis of response surface

Answer 1. The authors would like to thank the reviewere for the clarification. Suggested changes have been made in the manusript. 
1.    We have inserted what was previously Figure S1 into the manuscript as Figure 3
2.    We devided the Chapter 3 into subsection to clarify the main text, as suggested
3.    We described fitting the models into greater details (Lines 320-326)
4.    We compared our findindgs regarding the response surface results with findings of other authors (Lines 350-366)
5.    We dicussed response surface optimization in more depth (Lines 368-398)
6.    We additionally discussed our results regarding the impact of sample pretreatment on the yields and characteristics of pectin and compared it to available data regarding the pectin from other sources (since there are no such investigationst for tomato pectin) (Lines 496-507 and Lines 580-584).